# ASH2L Aggravates Fibrosis and Inflammation through HIPK2 in High Glucose-Induced Glomerular Mesangial Cells

**DOI:** 10.3390/genes13122244

**Published:** 2022-11-29

**Authors:** Wen Zhong, Chen Hong, Yejun Dong, Yuhui Li, Chenxi Xiao, Xinhua Liu

**Affiliations:** 1School of pharmacy, Fudan University, Shanghai 201203, China; 2Pharmacophenomics Laboratory, Human Phenome Institute, Fudan University, Shanghai 201203, China

**Keywords:** ASH2L, HIPK-2, high glucose, glomerular mesangial cells, diabetic nephropathy

## Abstract

Diabetic nephropathy (DN) is a leading cause of end-stage renal disease and continues to be a threat to patients with diabetes. Dysfunction of glomerular mesangial cells (GMCs) is the main contributing factor to glomerulosclerosis, which is a pathological feature of DN. The epigenetic factor ASH2L has long been thought to be a transcriptional activator, but its function and involvement in diabetic nephropathy is still unclear. Here, we investigated the effect of ASH2L on the regulation of fibrosis and inflammation induced by high glucose in mouse mesangial cells (mMCs). We observed that ASH2L expression is increased in high glucose-induced mMCs, while loss of ASH2L alleviated fibrosis and inflammation. Furthermore, ASH2L-mediates H3K4me3 of the homeodomain-interacting protein kinase 2 (HIPK2) promoter region, which is a contributor to fibrosis in the kidneys and promotes its transcriptional expression. Similar to loss of ASH2L, silencing HIPK2 also inhibited fibrosis and inflammation. In addition, ASH2L and HIPK2 are upregulated in the kidneys of both streptozocin-induced and db/db mouse. In conclusion, we uncovered the crucial role of ASH2L in high glucose-induced fibrosis and inflammation, suggesting that ASH2L regulation may be an attractive approach to attenuate the progression of DN.

## 1. Introduction

Diabetic nephropathy (DN) is the most common microvascular complication of diabetes and accounts for almost 50% of end-stage renal disease (ESRD) among patients with diabetes [1,2]. At least half of type 2 and one-third of type 1 diabetic patients develop DN during their lifetime [3]. Glomerular lesions are a pathological manifestation of DN [4]. Although hyperglycemia affects all types of glomerular cells in diabetic glomerulopathy, including endothelia cells, podocytes, and mesangial cells [5], the dysfunction of glomerular mesangial cells (GMCs) has crucial implications. Under high-glucose conditions, GMCs proliferate, exhibit hypertrophy, and secrete excessive extracellular matrix (ECM), creating an increased mesangial matrix and thickened glomerular basement membrane (GBM), resulting in glomerulosclerosis [5,6,7].

Studies have shown that GMCs lesions are multifactorial and involve crosstalk between inflammation and fibrosis [8]. Accumulation of ECM mainly containing collagen and fibronectin (FN) may cause interstitial fibrosis [9]. High glucose levels also activate chronic inflammation with inflammatory cells such as macrophages appearing to be associated with the progression of renal fibrosis [10]. Meanwhile, inflammation can induce collagen and FN expression, which increases ECM accumulation and accelerates glomerulosclerosis [11]. The nuclear factor-κB (NF-κB) signaling pathway is activated under hyperglycemic conditions along with the release of inflammatory chemokines, including TNF-α and IL-6 [12,13]. Current glycemic control strategies cannot effectively improve renal function and more work is needed to explore the pathological mechanism underlying DN [14].

Post-translational modifications of histone proteins, such as histone methylation, have been associated with fibrosis and inflammation [11]. ASH2L has been identified as a transcriptional activator in tumors and it plays a role in the regulation of H3K4 tri-methylation (H3K4me3). In endometrial cancer, it activates tumor cell proliferation and migration, and upregulates PAX2 (an oncogene of endometrial cancer) transcription [15]. Furthermore, the role of ASH2L has been demonstrated in blood cancer [16], pancreatic duct adenocarcinoma [17], and other tumors. However, it is not well understood whether and how ASH2L affects fibrosis and inflammation in DN.

HIPK2 (homeodomain-interacting protein kinase 2) was discovered more than a decade ago and is involved in a wide spectrum of biological functions [18]. By phosphorylating transcription factors, HIPK2 regulates the expression of related genes in cancer and nervous system diseases [19,20]. HIPK2 can phosphorylate and activate p53 to regulate apoptosis, and it can also regulate the p53-independent apoptosis pathway [21]. Additionally, HIPK2 has been implicated in renal fibrosis in recent studies [22]. Experimental and bioinformatics studies have identified HIPK2 as a key regulator of renal fibrosis, and both immunodeficient Tg26 mice and patients with various kidney diseases showed upregulated HIPK2 levels in the kidneys [23]. HIPK2 can activate p53, TGF-β/Smad3, Wnt/β-catenin, Notch, and NF-κB to regulate renal inflammation and fibrosis [24].

Our study demonstrated a novel epigenetic mechanism for ASH2L-mediated H3K4me3, which plays a key role in DN. High glucose levels induce ASH2L expression and H3K4me3 enrichment in the promoter region of HIPK2, which enhance HIPK2 expression and contribute to fibrosis and inflammation in DN.

## 2. Materials and Methods

### 2.1. Cell Culture and Reagents

SV40-MES-13 cells were obtained from Procell (Wuhan, China) and were cultured in low-glucose DMEM/F12 (3:1; *v*/*v*) containing 8% FBS (Gibco, New York, NY, USA) and 1% penicillin/streptomycin (BasalMedia, Shanghai, China) and grown at 37 °C in 5% CO_2_ incubator. To stimulate fibrosis and inflammation, the cells were treated with high glucose (11 mM, 22 mM or 33 mM) for 24 h, and mannitol was used to control osmotic pressure [25,26]. Specifically, the control group cells were given 5.5 mM D-glucose plus 27.5 mM mannitol; the 11 mM high glucose group cells were given 11 mM D-glucose plus 22 mM of mannitol; the 22 mM high glucose group cells were given 22 mM D-glucose and 11 mM of mannitol; and the 33 mM high glucose group cells were given 33 mM D-glucose without mannitol. Mannitol and D-glucose were obtained from Sigma-Aldrich (St. Louis, MO, USA). All other reagents were obtained from Sigma unless otherwise stated.

### 2.2. Small Interfering RNA (siRNA) Transfection

SV40-MES-13 cells in good growth conditions were seeded in six-well plates until approximately 60% confluency was achieved. 100 nM of control siRNA (siNC), ASH2L siRNA (si*Ash2l*), and HIPK2 siRNA (si*Hipk2*) were transfected using Lipofectamine RNA iMAX (Thermo Fisher Scientific, Waltham, MA, USA) according to the manufacturer’s instructions. The medium was replaced with DMEM/F12 (3:1) containing 2% FBS which was used for culturing for 12 h, followed by treatment with 33 mM high glucose (HG) for 24 h with mannitol regulating the osmotic pressure. The siRNA sequences used in this study are listed in Appendix A.

### 2.3. Knockdown Lentivirus Generation and Infection

Mouse pLVshRNA-U6-ASH2L-PGK-EGFP-E2A-Puro was obtained from GenePharma (Shanghai, China). This ASH2L knockdown plasmid was amplified according to the manufacturer’s instructions and transfected into 293T cells using recombinant PAX2 and PMD2.G vectors. 293T cells were cultured until the 72nd hour, and the cell culture media from the 48th and 72nd hour were mixed for later use. SV40-MES-13 cells were seeded in six-well plates until they reached 50% confluency. After filtering the virion with a 0.45 μm membrane, the virus solution was supplied to cells in the presence of 8 μg/mL polybrene for 24 h. The cells were collected after 48 h of culturing for further study.

### 2.4. Mice

The animal study protocol was approved by the Institutional Ethics Committee of Fudan University, China. Male mice (20–22 g) were allowed to fast for 12 h before injection and received an intraperitoneal injection of a low-dose of streptozocin (STZ, 50 mg/kg) for five consecutive days. One to two weeks after injection, the mice’s blood glucose was tested and selected as the model group (≥16.5 mmol/L). Male BKS-DB^(*Lepr*) WT/WT^ and BKS-DB^(*Lepr*) KO/KO^ mice (7–8 weeks old) were purchased from Jiangsu Jicuiyankang Biotechnology (Nanjing, China). The blood glucose levels of BKS-DB^(*Lepr*) KO/KO^ mice reached and sustained a high level (≥16.5 mmol/L) at the age of 8 weeks. The blood glucose levels of all animals were measured regularly, and they were harvested after 12 weeks. The mices’ left kidneys were dissected for Western blot analysis, and the right kidneys were subjected to histological analysis.

### 2.5. Western Blot Analysis

Immunoprecipitation-RIPA or RIPA lysis buffer containing 1% phosphatase inhibitor cocktail A and B, protease inhibitor cocktail (Biovision, Milpitas, CA, USA), and 1% PMSF (10 mM) were used to extract proteins from cells and tissues, respectively. To collect the total protein, the extractions were centrifuged at 12,000× *g* at 4 °C for 5 min. The protein concentration was estimated using a BCA kit (Thermo Fisher Scientific, Waltham, MA, USA), and the proteins were heated at 98 °C with LDS sample loading buffer and 5% β-mercaptoethanol. Equal amounts of protein samples were resolved by 10% SDS polyacrylamide gel electrophoresis and transferred onto an immunoblot nitrocellulose membrane (Millipore, Burlington, MA, USA). After blocking in 5% skim milk for 1–2 h at room temperature, the membranes were incubated with specific primary antibodies and horseradish peroxidase (HRP)-coupled secondary antibodies (Jackson ImmunoResearch Inc., West Grove, PA, USA), respectively. The blot findings were visualized with ECL detection reagent and a Bio-Rad Imager (Bio-Rad, Hercules, CA, USA). The primary antibodies used in this study are listed in Appendix A.

### 2.6. RNA Extraction and qRT-PCR

The total RNAs were extracted from cultured cells using TRIzol reagent (Takara Bio Inc., Dalian, China) according to a previously described method [27]. Six hundred ng of total RNA was reverse-transcribed into first-strand cDNA using the Prime RT Master Mix kit (Takara Bio Inc., Dalian, China). cDNA samples containing an equivalent amount of total RNA were quantified using SYBR Premix EX Taq II (Yeasen Biotech Co., Shanghai, China) with specific primer pairs and subjected to the CFX Connect™ Real-time PCR Detection System (Bio-Rad, Hercules, CA, USA). The specific primer pairs used are listed in Appendix A.

### 2.7. Immunofluorescence Analysis

Cells cultured on pretreated round coverslips were fixed in 4% paraformaldehyde, permeabilized in 0.1% Triton X-100, and then blocked in 10% goat serum. The cell slides were incubated with specific primary antibodies at 4 °C overnight, followed by incubation with Alexa Fluor-conjugated secondary antibodies (Invitrogen, Waltham, MA, USA), including goat anti-mouse IgG H&L (Alexa Fluor^®^ 488) and goat anti-rabbit IgG H&L (Alexa Fluor^®^ 594) (Invitrogen, Waltham, MA, USA). Cells were incubated briefly with or without 4’,6-diamidino-2-phenylindole (DAPI) for nuclear visualization and then imaged under a Zeiss fluorescence microscope (Carl Zeiss AG, Jena, Germany).

### 2.8. Mouse Kidney Histology Analysis

Mice were perfused with PBS, and samples from their right kidneys were fixed in 4% paraformaldehyde and embedded in paraffin. After deparaffinization and rehydration, the paraffin sections were stained with hematoxylin and eosin (H&E), Masson trichrome (Masson), and periodic acid–Schiff (PAS) using standard methods.

### 2.9. ChIP-PCR Analysis

Chromatin immunoprecipitation (ChIP) analysis was performed as per a previously described method [28]. Briefly, the cells were cross-linked with 1% formaldehyde and then terminated with 0.125 mM glycine. Nuclear lysates were sheared by sonication to generate DNA fragments, and DNA samples were isolated after immunoprecipitation with specific ASH2L and H3K4me3 antibodies. Normal rabbit IgG was used as the negative control. The DNA fragments were subjected to normal PCR using mouse HIPK2 promoter-specific primers: mus-HIPK2 0–500 bp forward: 5′-AGTCTTTAACCCTGCCCAGC-3′; reverse: 5′-CGAGGACCTGGGTTTGACTC-3′.

### 2.10. Statistical Analysis

All of the values are presented as the mean ± S.D. In vitro data were obtained from at least three biologically independent experiments and GraphPad Prism software was used to analyze the data and generate statistical graphs. The Student’s *t*-test was used for comparison between the two groups, and one-way ANOVA was used to analyze the differences between the multiple groups. The statistical significance between the groups was set at *p* < 0.05.

## 3. Results

### 3.1. High Glucose Activates ASH2L Aligned with Fibrosis and Inflammation in Mesangial Cells

To explore whether ASH2L regulates diabetic nephropathy, we used immortalized mMCs exposed to high glucose. Mesangial cells were treated with different concentrations of D-glucose (11 mM, 22 mM and 33 mM), with mannitol used as an osmotic control. High glucose conditions increased ASH2L expression (Figure 1A), which was accompanied by an increased expression of fibrosis markers (FN, Collagen 1, Collagen 3, and matrix metalloproteases-9 and -2 (MMP9 and MMP2)) and inflammatory mediators (iNOS, COX-2, IL-6, and IL-1β) in mMCs (Figure 1B,C). The 33 mM D-glucose (HG) treatment showed better performance than the other doses. Additionally, immunofluorescence (IF) assay detected a stronger fluorescence intensity of IL-6 in the high glucose group cells (Figure 1D), as well as enrichment of ASH2L and α-SMA (Figure 1E). Consistently, our results showed that HG treatment significantly upregulated mRNA levels of *Ash2l*, fibrosis-associated markers (*Col3a2*, *Mmp9*, and *Mmp2*), and inflammatory mediators (*Nos2* and *Il-6*) (Figure 1F). As ASH2L is mainly enriched in the nucleus, we excluded DAPI when stained for ASH2L for sake of reduced interference. Collectively, these results suggest that high ASH2L expression may contribute to fibrosis and inflammation in HG-induced renal mesangial cells.

### 3.2. Knockdown of ASH2L Decreases Fibrosis and Inflammation in Mesangial Cells

To further study the contribution of ASH2L to HG-induced fibrosis and inflammation, we transfected mMCs with ASH2L siRNA. The results showed that the loss of ASH2L decreased the high expression of fibrosis and inflammatory mediators (FN, Collagen 3, α-SMA, CTGF, iNOS, and IL-6) in HG-induced mMCs (Figure 2A). Consistently, further IF assay demonstrated that deletion of ASH2L lowered the fluorescence intensity of ASH2L, α-SMA, and IL-6 when compared to the HG group transfected with siNC (Figure 2B). The mRNA expression levels of *Nos2*, *Il-6*, and *Col3a2* were also downregulated by transfection with si*Ash2l* in HG-induced cells (Figure 2C). Furthermore, we transfected ASH2L shRNA with lentivirus to verify its function. Similarly, the knockdown of ASH2L with shRNA lentivirus decreased the elevated ASH2L expression induced by HG, along with the expression of inflammatory mediators (iNOS, COX-2, IL-6, and IL-1β) (Appendix A) and fibrosis markers (FN, collagen 3, collagen 1, MMP9, MMP2, and CTGF) (Appendix A). Additionally, the IF assay detected lower enrichment of ASH2L and α-SMA in HG group cells transfected with sh*Ash2l* than in the vector group (Appendix A). Taken together, the loss of ASH2L inhibited fibrosis and inflammation in HG-induced mMCs.

### 3.3. ASH2L-Mediacted H3K4me3 Promotes HIPK2 Transcription in High Glucose-Induced Mesangial Cells

As ASH2L is one of the WRAD complexes of H3K4 tri-methyltransferase [29], we subsequently verified the tri-methylation of H3K4. The results showed that the elevated levels of H3K4me3 in HG-induced cells were reversed by treatment with si*Ash2l* (Figure 3A). Considering that another H3K4 tri-methylation regulator may have also influenced the results, we conducted qRT-PCR to detect the mRNA levels of ASH2L and other tri-methylation regulators, such as KMT2B, KMT2D, and CXXC1. *Ash2l* was upregulated in HG-induced cells at different time points, but *Kmt2b*, *Kmt2d*, and *Cxxc1* were not. The *Ash2l* mRNA expression peaked after 3 h of treatment (Figure 3B). These results demonstrated that ASH2L-mediated H3K4me3 may play a special role in hyperglycemia-induced inflammation and fibrosis.

As HIPK2 has been identified as a key regulator of renal fibrosis [24], we determined whether ASH2L affects HIPK2 expression. We found that high glucose levels increased HIPK2 expression (Figure 3C,D). Considering that ASH2L is mainly located in the nucleus, we hypothesized that ASH2L could affect the gene expression of HIPK2 by regulating histone methylation. We conducted a ChIP assay using a specific ASH2L antibody. ChIP-PCR showed that ASH2L was enriched in the 0–500 bp promoter region of HIPK2 in HG-induced cells (Figure 3E). Furthermore, qRT-PCR was used to verify the transcription level of HIPK2, and the results showed that HG increased HIPK2 mRNA expression at the 12th hour of treatment (Figure 3F). This aligns with our previous findings where we detected elevated ASH2L mRNA levels at the 3rd hour of HG treatment (Figure 3C). In summary, these findings partially confirm our hypothesis that ASH2L may regulate H3K4me3 in HIPK2′s promoter region, which is responsible for triggering the transcription and expression of HIPK2.

### 3.4. Loss of ASH2L Suppresses HIPK2 Expression in High Glucose-Induced Mesangial Cells

We further verified our hypothesis on ASH2L’s effect on HIPK2 expression by delivering knockdown si*Ash2l* or sh*Ash2l* into cells. ChIP assay revealed that a loss of ASH2L diminished the enhanced binding of ASH2L and H3K4me3 to the 0–500 bp promoter region of HIPK2 in HG-induced cells (Figure 4A). qRT-PCR also demonstrated that treatment with si*Ash2l* downregulated the increased mRNA expression of HIPK2 induced by HG (Figure 4B). Furthermore, the loss of ASH2L decreased HIPK2 protein expression induced by HG (Figure 4C and Appendix A). Similarly, the IF assay showed direct visualization of dimeric HIPK2 after si*Ash2l* or sh*Ash2l* treatment in HG-induced cells (Figure 4D and Appendix A). In summary, we demonstrated for the first time that ASH2L activates HIPK2 transcription and increases its protein expression through tri-methylation of H3K4 histones in the 0–500 bp fragment of the *Hipk2* promoter region, thus mediating fibrosis and inflammation in HG-induced mMCs.

### 3.5. Loss of HIPK2 Decreases Fibrosis and Inflammation in Mesangial Cells

After identifying the regulatory function and mechanism of ASH2L in HIPK2 expression, we verified the role of HIPK2 in fibrosis and inflammation induced by high glucose. qRT-PCR quantification determined that si*Hipk2* lowered *Hipk2* mRNA expression induced by HG (Figure 5A). Additionally, si*Hipk2* treatment decreased the levels of inflammatory mediators (IL-6 and IL-1β) (Figure 5B) and fibrosis-related proteins (FN, Collagen 3, Collagen 1, MMP9, α-SMA, and CTGF) (Figure 5C) in HG-induced mMCs. The IF assay proved that IL-6 and α-SMA were downregulated by si*Hipk2* (Figure 5D,E). These results demonstrate that HIPK2 can serve as a downstream determinant of ASH2L, and it plays a crucial role in regulating fibrosis and inflammation in HG-induced mMCs.

### 3.6. Diabetic Mice Show Renal Injury and Upregulated ASH2L and HIPK2 Allied with Fibrosis and Inflammation

Both STZ-induced diabetic mice and genetically defective db/db mice were used as DN models. The expression of ASH2L, HIPK2, and H3K4me3 was upregulated in STZ-induced type 1 diabetic (T1D) mice and type 2 diabetic (T2D) db/db mice (Figure 6A). Similarly, fibrosis- and inflammation-related proteins were also increased in the kidney tissues of both STZ-induced and db/db mice (Figure 6B). Both STZ-induced diabetic mice and db/db mice showed sustained hyperglycemia (Figure 6C). Additionally, histological assessment of their kidneys revealed some typical lesions with thickening of the glomerular basement membrane (indicated by red arrows) and cell hyperplasia in capillaries (indicated by black arrows) in H&E staining, increased PAS-positive areas with PAS staining, and more deposition of collagen with Masson’s staining (Figure 3D). These results indicate renal injury in diabetic mice, and that the ASH2L-HIPK2 axis partially regulates fibrosis and inflammation in DN.

## 4. Discussion

In this study, we identified that the epigenetic regulator ASH2L promotes fibrosis and inflammation in HG-treated glomerular mesangial cells. ASH2L could activate the transcription of HIPK2 (a key regulator of renal fibrosis) and increase its protein expression through H3K4me3 in the HIPK2 promoter region, thereby mediating renal fibrosis and inflammation in DN. In contrast, loss of ASH2L or HIPK2 blocked the activation of fibrosis and inflammation in HG-treated mMCs, suggesting that ASH2L may be a promising interference target for DN.

DN is one of the microvascular complications of diabetes [3] and brings a great economic and personal burden to patients [30]. Glomerulopathy is defined as the earliest and most critical lesion factor in DN [31] and is usually related to the coeffects of inflammation and fibrosis. The presence of chronic hyperglycemia can cause the persistence of the local inflammatory response and accumulation of ECM in renal tissue [32]. Excess collagen and FN mainly secreted by mesangial cells are also packed into the mesangial matrix, which leads to the expansion and sclerosis of the mesangial matrix [33]. Inflammatory cytokines, including TNF-α, IL-6, MCP-1, ICAM-1, CSF-1, and MyD88, were detected to be elevated in serum or peripheral blood cells in DN, which increased with disease progression [34]. Among intrinsic glomerular cells, GMCs have been correlated with the severity of albuminuria in DN [35]. Under high glucose conditions, GMCs are stimulated, and they secrete ECM to promote glomerular fibrosis [36]. Meanwhile, there is also cross talk between fibrosis and inflammation in GMCs lesions, and the progression of renal fibrosis is closely associated with the inflammatory response [37]. Inflammatory cells can induce fibrosis and regulate matrix turnover by producing MMPs [38]. Inflammatory cytokines secreted by immune cells also act as stimuli for GMCs [39]. Recent research has found that renal macrophages are closely related to the accumulation of renal interstitial matrix protein and the degree of interstitial fibrosis in GMCs, which may be related to the secretion of IL-1β and TNF-α [40]. Our study also confirmed that the upregulated expression of fibrosis and inflammatory mediators was detectable in mMCs induced by high glucose.

Epidemiological studies have shown that improvements in diabetes management in recent decades have reduced the incidence of diabetes-related cardiovascular diseases but have had only a minimal impact on the incidence of ESRD [41]. Therefore, it is important to reflect on the pathogenesis of DN to develop new prevention and treatment strategies. We found that the protein expression and mRNA levels of ASH2L were significantly upregulated in mMCs induced by HG and accompanied by an activation of fibrosis and inflammation in vitro. In contrast, the loss of ASH2L inhibited HG-induced ASH2L activation and attenuated fibrosis and inflammation in mMCs. These results indicate that HG conditions activate fibrosis and inflammation in mesangial cells, causing GMCs cytopathies, and ASH2L may play a crucial role in this process.

Interestingly, we discovered that ASH2L was mainly expressed in the nucleus in the fluorescence image, which suggests that ASH2L is typically identified as a transcription igniter. ASH2L is one of the components of the WRAD complex, which is a core component of histone methyltransferase MLL, and comprises full-length WDR5, RBBP5, ASH2L, and DPY30 (WRAD) proteins and MLL1 [29,42]. Researchers found that ASH2L is required for H3K4 trimethylation and the loss of ASH2L reduced H3K4 trimethylation, but with reduced effects on H3K4 mono- and dimethylation levels [43,44]. ASH2L can activate the transcription of downstream genes by promoting H3K4me3 [45], and high levels of H3K4me3 are correlated with open chromatin structure and active transcription [46]. As other factors, such as KMT2B, KMT2D, and CXXC1, can also regulate H3K4me3, we analyzed their expressions in mMCs. We found that the mRNA expressions of *Kmt2b*, *Kmt2d*, and *Cxxc1* did not change significantly in HG-induced cells, indicating that ASH2L-mediated H3K4me3 may have a special biological function in DN.

However, only a few studies have focused on ASH2L, and its regulatory role in DN has not received much attention. After demonstrating that ASH2L contributes to fibrosis and inflammation, we explored the mechanism by which ASH2L regulates this fibrosis and inflammation. Our study found that the protein kinase HIPK2 may be a downstream target of ASH2L. HIPK2 is considered a tumor suppressor gene that mediates the activation of Wnt, Notch, and TGF-β-induced signaling pathways [19]; however, recent studies have focused on its regulatory role in the kidneys and other tissues [24]. Studies have shown that HIPK2 protein is highly expressed in the kidneys of STZ-induced DN mice and in mesangial cells induced by high glucose [20], which is consistent with our study. Using ChIP-PCR, we further verified that ASH2L could bind to the promoter region of HIPK2 and activate its transcriptional expression. In contrast, the loss of ASH2L alleviated the high level of H3K4me3 in the promoter region of HIPK2 found in mMCs induced by HG. Similarly, we found that silencing HIPK2 also reduced the high fibrosis and inflammation protein expression which was induced by HG. Consistent with the in vitro findings, ASH2L and HIPK2 expression in the kidney tissues of STZ-induced diabetic mice and db/db mice increased, along with elevated levels of fibrosis and inflammation-related proteins. These results indicate that ASH2L triggers HIPK2 transcription by regulating H3K4me3 on its promoter region and thus may play a crucial role in fibrosis and inflammation under high glucose conditions.

Currently, studies on HIPK2 have expanded to include a wider range of fields. Donninger et al. identified that HIPK2-mediated p53 is critical and specific for Ras-mediated senescence through post-translational modifications [47]; HIPK2 is also involved in cell survival as a regulator of DNA damage response through p53 [48]. Researchers have also reported that exercise attenuates cardiomyocyte apoptosis and protects against myocardial infarction by suppressing HIPK2 [49]. These findings suggest that HIPK2 has broad research potential. The role of HIPK2 in renal fibrosis has been confirmed in mice with unilateral ureteral obstruction and mice with acute kidney injury [22,50]. Although the upregulation of HIPK2 in the kidneys of STZ-induced mice and db/db mice has been observed in this study and in other studies, further exploration is needed to investigate how HIPK2 affects DN. Specifically, whether HIPK2 works through the proven TGF- β1/Smad3 [51], EGFR/STAT3 [50], or NF-κB signaling pathway [52].

Hence, one of the limitations of this study was the failure to further explore the specific mechanisms by which HIPK2 affects DN. Additionally, we conducted prime ChIP-PCR to detect the abundance of ASH2L and H3K4me3 binding to the *Hipk2* promoter, which could not elucidate the specific *Ash2l* binding motifs. Moreover, because of the specific activation of ASH2L and HIPK2 in DN, further studies are needed to explore their inhibitors as a potential therapeutic approach to attenuate the progression of DN.

## 5. Conclusions

In summary, this study revealed the first instance of the epigenetic regulator ASH2L participating in high glucose-induced fibrosis and inflammation through transcriptional activation of HIPK2, suggesting that there is potential for attenuate the progression of DN.

## Figures and Tables

**Figure 1 genes-13-02244-f001:**
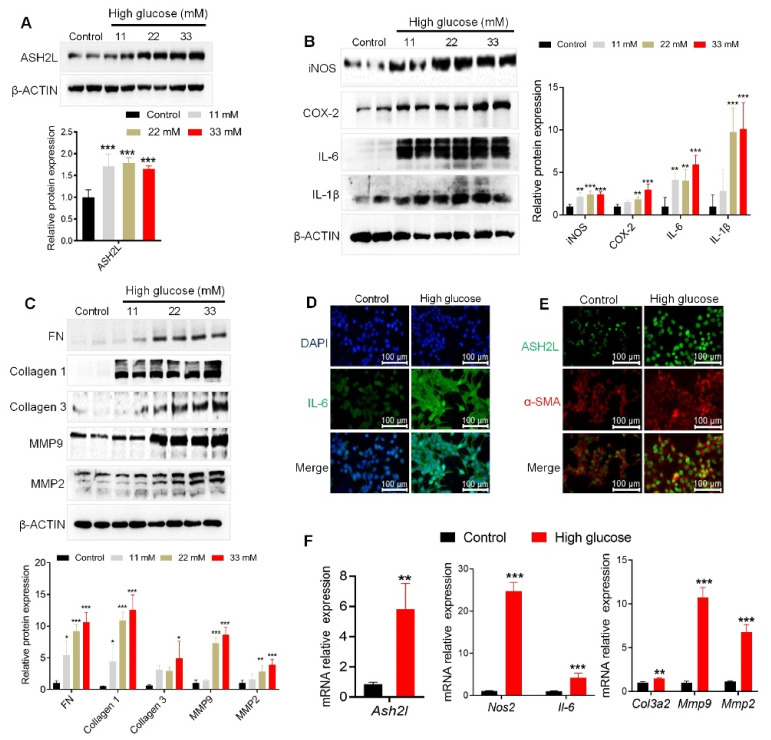
High glucose activates ASH2L aligned with fibrosis and inflammation in mesangial cells. (**A**–**C**) Western blot analysis of ASH2L, fibrosis markers (FN, Collagen 1, Collagen 3, MMP9, and MMP2), and inflammatory mediators (iNOS, COX-2, IL-6, and IL-1β) expression in SV40-MES-13 cells treated with 11 mM, 22 mM, and 33 mM D-glucose for 24 h. Data from at least three independent experiments are shown as mean ± S.D, * *p* < 0.05, ** *p* < 0.01, and *** *p* < 0.001 compared with the control group. (**D**,**E**) Immunofluorescence staining of IL-6, ASH2L, and α-SMA in SV40-MES-13 cells treated with or without 33 mM D-glucose (high glucose) for 24 h. (**F**) Quantitative RT-PCR analyses of the relative mRNA levels of *Ash2l*, fibrosis-associated markers (*Col3a2*, *Mmp9*, and *Mmp2*), and inflammatory mediators (*Nos2* and *Il-6*) in SV40-MES-13 cells treated with or without 33 mM D-glucose. Data from at least three independent experiments are shown as mean ± S.D, ** *p* < 0.01, and *** *p* < 0.001 compared with the control group.

**Figure 2 genes-13-02244-f002:**
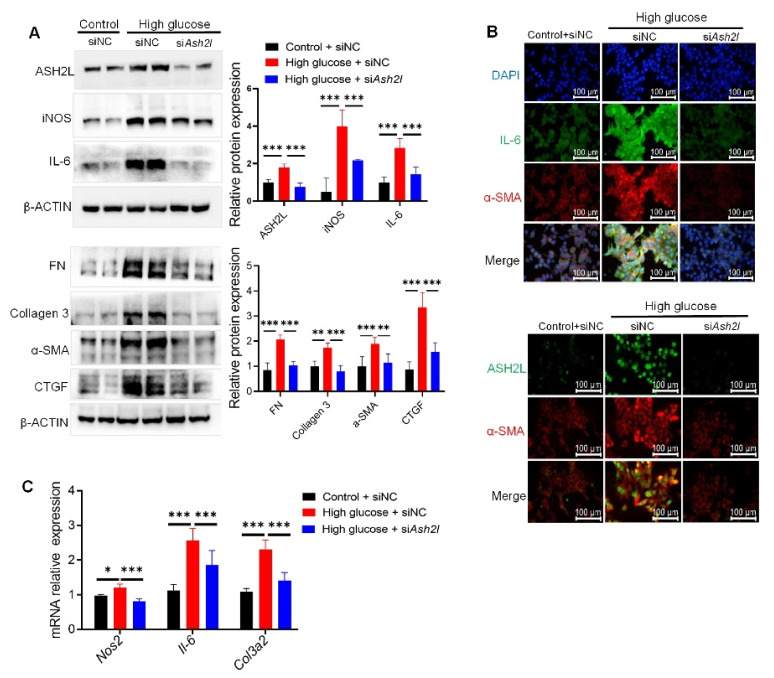
Knockdown of ASH2L decreases fibrosis and inflammation in mesangial cells. (**A**) Western blot analysis of ASH2L, fibrosis markers (FN, Collagen 3, α-SMA, and CTGF), and inflammatory mediators (iNOS, and IL-6) expression in SV40-MES-13 cells treated with or without 33 mM D-glucose for 24 h after being transfected with negative control (siNC) or ASH2L small interfering RNA (si*Ash2l*). Data from at least three independent experiments are shown as mean ± S.D, ** *p* < 0.01, and *** *p* < 0.001. (**B**) Immunofluorescence staining of ASH2L, α-SMA, and IL-6 in SV40-MES-13 cells treated with or without 33 mM D-glucose for 24 h after transfection with siNC or si*Ash2l*. (**C**) Quantitative RT-PCR analysis of *Nos2, Il-6,* and *Col3a2* in SV40-MES-13 cells treated with or without 33 mM D-glucose for 24 h after transfection with siNC or si*Ash2l*. Data from at least three independent experiments are shown as mean ± S.D, * *p* < 0.05, and *** *p* < 0.001.

**Figure 3 genes-13-02244-f003:**
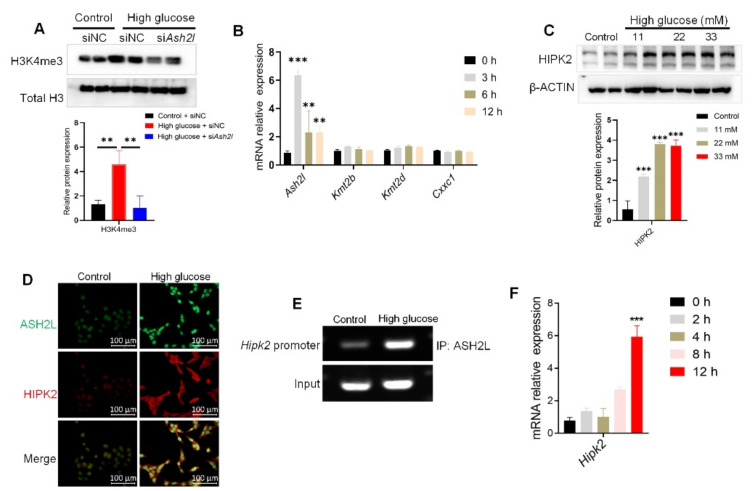
ASH2L-mediated H3K4me3 promotes HIPK2 transcription in high glucose-induced mesangial cells. *(***A**) Western blot analysis of H3K4me3 in SV40-MES-13 cells treated with or without 33 mM high glucose for 24 h after being transfected with negative control (siNC) or ASH2L small interfering RNA (si*Ash2l*). Data from at least three independent experiments are shown as mean ± S.D and ** *p* < 0.01. (**B**) Quantitative RT-PCR analyses of *Ash2l*, *Kmt2b*, *Kmt2d,* and *Cxxc1* in SV40-MES-13 cells treated with 33 mM D-glucose for 0, 3, 6, and 12 h. Data from at least three independent experiments are shown as mean ± S.D, ** *p* < 0.01, and *** *p* < 0.001 compared with the control group. (**C**) Western blot analysis of HIPK2 expression in SV40-MES-13 cells treated with 11 mM, 22 mM, and 33 mM high glucose for 24 h. Data from at least three independent experiments are shown as mean ± S.D and *** *p* < 0.001 compared with the control group. (**D**) Immunofluorescence staining of ASH2L and HIPK2 in SV40-MES-13 cells treated with or without 33 mM D-glucose for 24 h. (**E**) ChIP-PCR assay of the 0–500 bp promoter region of HIPK2 in SV40-MES-13 cells treated with 33 mM high glucose for 24 h. (**F**) Quantitative RT-PCR analyses of *Hipk2* in SV40-MES-13 cells treated with 33 mM high glucose for 0, 2, 4, 8, and 12 h. Data are shown as mean ± S.D and *** *p* < 0.001 compared with the control group.

**Figure 4 genes-13-02244-f004:**
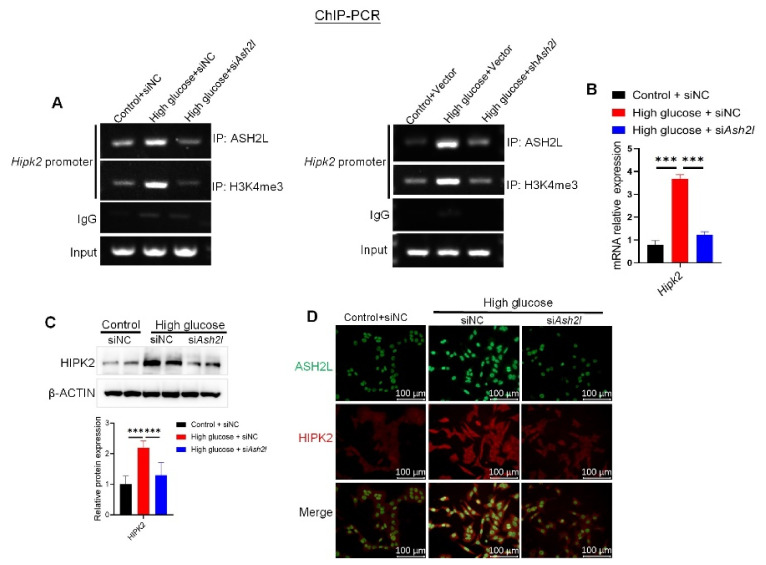
Loss of ASH2L suppresses HIPK2 expression in high glucose induced mesangial cells. (**A**) ChIP analysis of ASH2L binding to HIPK2 gene promoter in SV40-MES-13 cells. (**B**) Quantitative RT-PCR analyses of *HIPK2* in SV40-MES-13 cells treated with or without 33 mM D-glucose for 24 h after transfection with negative control (siNC) or ASH2L small interfering RNA (si*Ash2l*). Data are shown as mean ± S.D from at least three independent experiments with *** *p* < 0.001. (**C**) Western blot analysis of HIPK2 expression in SV40-MES-13 cells. Data are shown as mean ± S.D from at least three independent experiments and *** *p* < 0.001. (**D**) Immunofluorescence staining of ASH2L and HIPK2 in SV40-MES-13 cells.

**Figure 5 genes-13-02244-f005:**
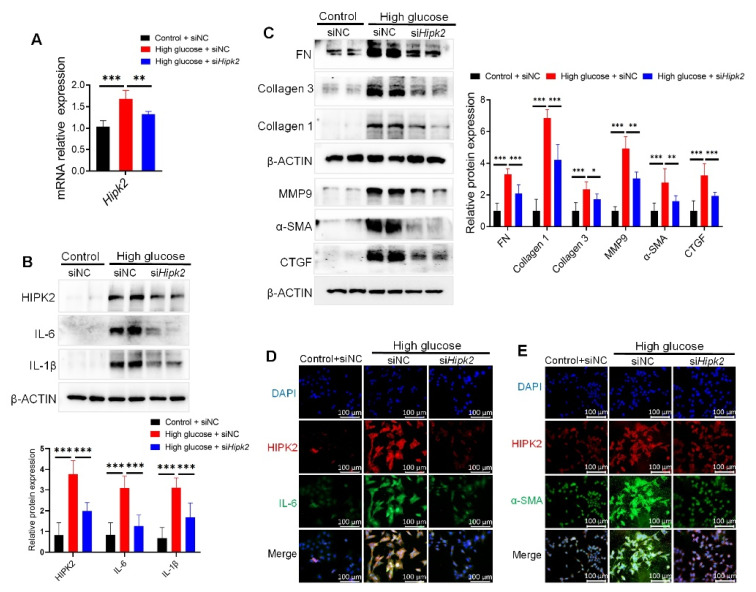
Loss of HIPK2 decreases fibrosis and inflammation in mesangial cells. (**A**) Quantitative RT-PCR analyses of *Hipk2* in SV40-MES-13 cells treated with or without 33 mM high glucose for 24 h after transfection with negative control (siNC) or HIPK2 small interfering RNA (si*Hipk2*). Data from at least three independent experiments are shown as mean ± S.D, ** *p* < 0.01, and *** *p* < 0.001. (**B**,**C**) Western blot analysis of HIPK2, inflammatory mediators (IL-6 and IL-1β), and fibrosis markers (FN, Collagen 3, Collagen 1, MMP9, α-SMA, and CTGF) expression in SV40-MES-13 cells treated with or without 33 mM high glucose for 24 h after transfection with siNC or si*Hipk2*. Data from at least three independent experiments are shown as mean ± S.D, * *p* < 0.05, ** *p* < 0.01 and *** *p* < 0.001. (**D**,**E**) Immunofluorescence staining of HIPK2, α-SMA, and IL-6 in SV40-MES-13 cells treated with or without 33 mM D-glucose for 24 h after transfection with siNC or si*Hipk2*.

**Figure 6 genes-13-02244-f006:**
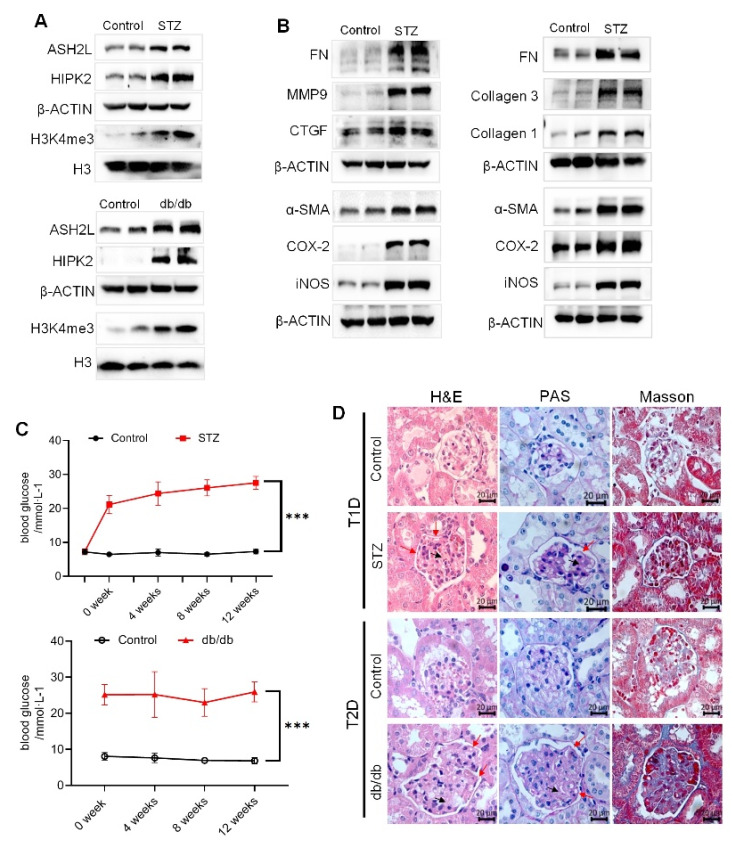
Diabetic mice show renal injury and upregulated ASH2L and HIPK2 allied with fibrosis and inflammation. (**A**) Western blot analysis of ASH2L, HIPK2, and H3K4me3 in STZ-induced mice (*n* = 5) and db/db mice (*n* = 3). (**B**) Western blot analysis of fibrosis markers and inflammatory markers in STZ-induced mice (*n* = 5) and db/db mice (*n* = 3). (**C**) Blood glucose of STZ-induced mice (*n* = 10) and db/db mice (*n* = 6). Data are shown as mean ± S.D and *** *p* < 0.001. (**D**) Representative images of H&E staining, PAS staining, and Masson’s staining of mice kidneys.

## Data Availability

Not applicable.

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
