# Peer review of "ASH2L Aggravates Fibrosis and Inflammation through HIPK2 in High Glucose-Induced Glomerular Mesangial Cells"

_genes, 2022, doi:10.3390/genes13122244_

Round 1
Reviewer 1 Report
The authors analyzed the role of Ash2l and its downstream target Hipk2 for mesangial fibrosis and inflammation in the context of high glucose and diabetes. Regarding the growing number of patiens suffering from diabetic kidney disease, the data presented are meaningful. The design of the study is well structured and comprehensible. Despite the very intersting approach, the manuscript should be improved to increase significance.
The resolution of the figures should be optimized, especially for immunofluorescence stainings. In figure 1E, the scale bar is missing. In all other figures showing immunofluorescence stainings, the caption of the scale bar is not readable.
Referring to the subordinate clause in line 213: what does this mean “ASH2L is one of the WARD complexes”? (To my knowledge Ash2l is part of the WARD complex.) What is the WARD complex? Please, specify the stated expression and give a short definition of WARD either here or in the introduction.
In line 218 “Ash2l, but not Kmt2b, Kmt2b, nor Cxxc1”: Did you mean “… not Kmt2b, Kmt2d, nor…”?
A strong effect of siASH2L under hyperglycemia was demonstrated in figure 2-5. Is there also an effect on the respective expression after silencing SDJ2L under normoglycemic conditions?
While figure 1 and 2 display changes in expressions after 24h of hyperglycemia, figure 2B shows expression of Ash2l, Kmt2b, Kmt2d, and Cxxc1 over the time from time point zero until 12h after induction. I would appreciate if the RNA expression of these genes could also be shown after 24h. Additionally, the authors should also analyze the expression of these genes on protein level or potential changes in transcriptional activity.
Ash2l can bind DNA, binding motifs are published (eg: DOI: 10.1074/jbc.M112.424515). Can these (or other) Ash2l binding motifs be found in the promoter region of Hipk2? Additionally, promoter activity assays would further validate the direct control of Hipk2 by Ash2l.
The caption of figure 4A right is not complete. Does figure 4A show the result of a PCR (ChIP PCR?) or an immunoblot (the heading “ChIP-PCR” (or at least the position of the heading “ChIP-PCR” is confusing)?
Figure 6: There is only a western blot for H3K4me3 and H3 in db/db mice, not STZ mice. Please, add the missing blots.
The limitations of this study should be discussed.
Author Response
The pictures cannot be displayed on the website reply box, so please see the attachment.

Reviewer 2 Report
Interesting and innovative work, which directly correlates, perhaps for the first time, the glomerular damage with the increase of ASH2L, which would regulate the tri-methylation of the HIPK2 promotor, with consequent fibrosis. While the results supporting the results appear solid, there are some points to improve for a better understanding of the text. In general, English needs revision by the authors.
· Materials and methods.
In paragraph 2.1 there is no reference to the origin of the cells
· The introduction
I would like more insight into the function of tri-methylation in transcription. Furthermore, the WRAD complex of which ASH2L is part and its function is never mentioned in the introduction. The role of HIPK2 appears ambiguous (lines 52-61): does it inhibit or activate p53?
Many important concepts are reported in the discussion. The two parts could be better balanced in order to introduce the article comprehensively.
· Below are some specific parts that need revision:
Title: “high glucose induced” add dash between “glucose
Line 15-17: It is not clearly understood; especially the use of the verb "medicates".
Line 17: “Similarly” to what?
Line 25: Usually the acronym for “end stage renal disease” is ESRD, not ERD.
Line 30-33: The main verb of the sentence is missing, in necessary to rephrase.
Line 47-48: repetition of “endometrial cancer”.
Line 52-61: Does HIPK2 inhibit or activate p53? Moreover, p53 is a tumor suppressor, while the word “onco-factor” seems similar to oncogene.
Line 215-217: Rephrase, please.
Author Response
- Materials and methods.
- In paragraph 2.1 there is no reference to the origin of the cells
A: Thanks for your important amendment. SV40-MES-13 cells were obtained from Procell Life Science&Technology (Wuhan, China), and we have elaborated this information in the manuscript.
- The introduction
- I would like more insight into the function of tri-methylation in transcription. Furthermore, the WRAD complex of which ASH2L is part and its function is never mentioned in the introduction. The role of HIPK2 appears ambiguous (lines 52-61): does it inhibit or activate p53?
A: Thank you for your comments. In general, monomethylation, dimethylation, or trimethylation (H3K4me1/2/3) of the histone H3 lysine 4 are enriched in transcriptionally active genomic regions, and H3K4me3 is generally associated with activated gene transcription (DOI: 10.1016/j.cell.2007.02.005; DOI: 10.1016/j.tcb.2011.11.001).
Besides, thanks for your suggestion on ASH2L and WARD complex, and we have given a short expression for ASH2L and WARD complex in the discussion. ASH2L is one of the components of the WRAD complex, and WRAD complex is a core component of histone methyl-transferase MLL, which is composed of full-length WDR5, RBBP5, ASH2L and DPY30 (WRAD) proteins and MLL1 (DOI: 10.1073/pnas.1109360108).
As to the role of HIPK2 on p53, p53 is an important tumor suppressor gene, and HIPK2 acts as a protein kinase and can phosphorylate transcription factors to control gene expression. It was found that HIPK2 can bind to and phosphorylate p53, improve p53 transcriptional activity, and increase the expression stability of p53 protein (DOI: 10.1186/1476-4598-8-85). Here, thank you for pointing out the problem, which has been modified in the manuscript.
- Many important concepts are reported in the discussion. The two parts could be better balanced in order to introduce the article comprehensively.
A: Thanks for your valuable suggestion. The concepts have been readjusted, and we really hope that the flow and comprehensiveness have been improved.
- Below are some specific parts that need revision:
- Title: “high glucose induced” add dash between “glucose
A: Grateful for your suggestion, a dash has been added between "high glucose" and "induced" in the title.
- Line 15-17: It is not clearly understood; especially the use of the verb "medicates".
A: Thanks for pointing out the language problems. This sentence has been improved.
- Line 17: “Similarly” to what?
A: Thank you for pointing out the vagueness, “Similarly” here means that the effect of siHipk2 treatment was similar to the effects of siAsh2l—silencing ASH2L or HIPK2 both can inhibit fibrosis and inflammation induced by high glucose in cells, and the word has been replaced by “Similar to loss of ASH2L” to avoid the vagueness.
- Line 25: Usually the acronym for “end stage renal disease” is ESRD, not ERD.
A: Grateful for your important amendment. The miss used acronym has been replaced by “ESRD”.
- Line 30-33: The main verb of the sentence is missing, in necessary to rephrase.
A: Appreciate for your advice, the sentence has been reworded.
- Line 47-48: repetition of “endometrial cancer”.
A: Thanks for pointing out the repetition, the repeated “endometrial cancer” has been deleted.
- Line 52-61: Does HIPK2 inhibit or activate p53? Moreover, p53 is a tumor suppressor, while the word “onco-factor” seems similar to oncogene.
A: Sincere thanks for your comments. As you kindly reminded, p53 is an important tumor suppressor, and HIPK2 can activate p53, which is related to modifications at Ser46 of p53 (DOI: 10.1186/1476-4598-8-85). Activation of p53 by HIPK2 was crucial for the transcription of pre-apoptotic genes and induction of apoptosis, which makes HIPK2 a tumor suppressor too. In this manuscript, we focused more on the role of HIPK2 in kidney fibrosis, and we would truly thank you for pointing out the misuses.
- Line 215-217: Rephrase, please.
A: Grateful for your advice, the expression has been improved to be more concise. The authors feel apologetic for the inappropriate language of this manuscript and are willing to improve the text based on your valuable comments and beyond.
Reviewer 3 Report
Glomerular mesangial cell (GMC) is a major contributor to glomerulosclerosis. This study reports the role of epigenetic factor ASH2L in fibrosis and inflammation through HIPK2 in high glucose-induced mice GMC. Loss of ASH2L is reported to alleviate fibrosis and inflammation in a process mediated by HIPK2, a contributor to renal fibrosis. Both, ASH2L and HIPK2 were up-regulated in STZ and db/db mouse, and silencing HIPK2 inhibited fibrosis and also inflammation markers, indicating the potential role of ASH2L as a therapeutic target in diabetic nephropathy.
Comments,
In the methods section, Line 71, high glucose is reported to stimulate “fibrosis and inflammation”. Reference should be provided in support. Also, the experimental details need to be elaborated.
The sentence structure in Line 85, “amplificated as the manufacturer’s guidance described…” is awkward. Please improve. Similarly, Line 107, “from cells and issues”, Line 144, “then were immunoprecipitation with”, Line 152, “student's T test” (or Student's t-test!), Line 161, “was companied by” needs to be improved.
In the manuscript, authors have used the words, “increased fibrosis” in context of results in cell culture. Fibrosis is a property of tissue. In cell culture, at best it can be said fibrosis markers.
Results provide evidence partially supporting the ASH2L and HIPK2 expression in renal fibrosis in high glucose induced glomerular mesangial cells via HIPK2 through H3K4me3. What are the limitations of the study?
The study (discussion part) fails to correlate fibrosis and inflammatory chemokines, cytokines, even though it has a passing remark in Line 328-329. The discussion part needs to be more mechanistic.
Authors may reconsider the use of word ‘Generally’ in the Conclusions. Also, adding a line on cross talk b/w fibrosis and inflammatory mediators may add value. The use of word ‘manipulate’ (Line 384) does not sound good.
Overall, the manuscript is technically fine but has language issues. For example, “ASH2L medicates H3K4me3 of the promotor region” (Line 15). Similarly, Line 31 (“GMCs proliferation, hypertrophy, and secrete excessive …”), Line 68 (“cultured in the medium that mixed with low glucose”), Line 170, “ASH2L may contributes to”, and the language at many other places throughout the text need to be improved.
Author Response
Comments,
- In the methods section, Line 71, high glucose is reported to stimulate “fibrosis and inflammation”. Reference should be provided in support. Also, the experimental details need to be elaborated.
A: Thank you for your advice, we have elaborated the experimental details, and provided references related to the cell culture of SV40-MES-13 cells.
- The sentence structure in Line 85, “amplificated as the manufacturer’s guidance described…” is awkward. Please improve. Similarly, Line 107, “from cells and issues”, Line 144, “then were immunoprecipitation with”, Line 152, “student's T test” (or Student's t-test!), Line 161, “was companied by” needs to be improved.
A: Thanks for pointing out the language problems, the authors have checked and corrected all the mistakes and inappropriateness.
- In the manuscript, authors have used the words, “increased fibrosis” in context of results in cell culture. Fibrosis is a property of tissue. In cell culture, at best it can be said fibrosis markers.
A: Thanks for the important amendment, we have checked and changed the articulation to “fibrosis markers” or “fibrosis-related proteins”.
- Results provide evidence partially supporting the ASH2L and HIPK2 expression in renal fibrosis in high glucose induced glomerular mesangial cells via HIPK2 through H3K4me3. What are the limitations of the study?
A: Thanks for your comment. One of the limitations of this study was that the ChIP-PCR we conducted to detect the abundance of ASH2L and H3K4me3 binding to the Hipk2 promoter could not elucidate the specific ASH2L binding motifs. Moreover, this study failed to explore further on the specific mechanisms by which HIPK2 affects DN.
- The study (discussion part) fails to correlate fibrosis and inflammatory chemokines, cytokines, even though it has a passing remark in Line 328-329. The discussion part needs to be more mechanistic.
A: Grateful for your suggestion, we have supplemented further materials on the relationship between fibrosis and inflammatory chemokines to the discussion part, and referenced to recent research that found renal macrophages are closely related to the accumulation of renal interstitial matrix protein and the degree of interstitial fibrosis in GMCs, which may relate to the secreted IL-1β and TNF-α (Doi:10.3389/fimmu.2022.835879).
- Authors may reconsider the use of word ‘Generally’ in the Conclusions.
A: Thanks for identifying the misnomer, we have modified it to “In summary”.
- Also, adding a line on cross talk b/w fibrosis and inflammatory mediators may add value.
A: Appreciate for your advice, we have added this sentence in our paragraph and revised this part of the description.
- The use of word ‘manipulate’ (Line 384) does not sound good.
A: Thank you for pointing out the misnomer, we have improved this sentence.
- Overall, the manuscript is technically fine but has language issues. For example, “ASH2L medicates H3K4me3 of the promotor region” (Line 15). Similarly, Line 31 (“GMCs proliferation, hypertrophy, and secrete excessive …”), Line 68 (“cultured in the medium that mixed with low glucose”), Line 170, “ASH2L may contributes to”, and the language at many other places throughout the text need to be improved.
A: Heartfelt thanks for your comments, these sentences have been improved according to your kind advice. The authors feel apologetic for the awkward language of this manuscript and are meanwhile inspired to improve the text based on your helpful comments and beyond. We have now sought professional help for language corrections. We really hope that the flow and language level have been substantially improved.
Round 2
Reviewer 1 Report
The authors strongly improved their manuscript. Now, the introduction provides a more precise lead into the topic. The material and methods section is rich in detail. The results are presented more clearly andthe discussion has improved substantially.
I am fine with the new version.